# Nonparametric Limits of Agreement in Method Comparison Studies: A Simulation Study on Extreme Quantile Estimation

**DOI:** 10.3390/ijerph17228330

**Published:** 2020-11-11

**Authors:** Oke Gerke

**Affiliations:** 1Department of Nuclear Medicine, Odense University Hospital, 5000 Odense C, Denmark; oke.gerke@rsyd.dk; 2Department of Clinical Research, University of Southern Denmark, 5000 Odense C, Denmark

**Keywords:** agreement, Bland–Altman analysis, coverage, limits of agreement, method comparison, quantile estimation, repeatability, reproducibility

## Abstract

Bland–Altman limits of agreement and the underlying plot are a well-established means in method comparison studies on quantitative outcomes. Normally distributed paired differences, a constant bias, and variance homogeneity across the measurement range are implicit assumptions to this end. Whenever these assumptions are not fully met and cannot be remedied by an appropriate transformation of the data or the application of a regression approach, the 2.5% and 97.5% quantiles of the differences have to be estimated nonparametrically. Earlier, a simple Sample Quantile (SQ) estimator (a weighted average of the observations closest to the target quantile), the Harrell–Davis estimator (HD), and estimators of the Sfakianakis–Verginis type (SV) outperformed 10 other quantile estimators in terms of mean coverage for the next observation in a simulation study, based on sample sizes between 30 and 150. Here, we investigate the variability of the coverage probability of these three and another three promising nonparametric quantile estimators with n=50(50)200,250(250)1000. The SQ estimator outperformed the HD and SV estimators for n=50 and was slightly better for n=100, whereas the SQ, HD, and SV estimators performed identically well for n≥150. The similarity of the boxplots for the SQ estimator across both distributions and sample sizes was striking.

## 1. Introduction

When comparing methods for the measurement of a continuous outcome, the Bland–Altman Limits of Agreement (BA LoAs) are a well-known and well-established means to this end [1,2,3]. Mean values are plotted against the paired differences in a scatter plot that is supplemented by an estimate of the bias (i.e., the mean of the paired differences) and the so-called LoAs (bias estimate +/− 1.96 standard deviations of the paired differences), including the respective 95% confidence intervals [4,5,6,7]. Under the assumptions of normally distributed paired differences, the BA LoAs represent estimates for the boundaries between which 95% of all population differences are supposed to lie. In the case of the variance heterogeneity of the paired differences, a normalizing transformation (like the natural logarithm) may render the usual analysis (on the log-scale) possible, whereas a bias that is non-constant across the measurement range might require a regression approach [3,8]. If neither a transformation of the data, nor a regression approach are applicable, the LoAs are often estimated by simple empirical 2.5% and 97.5% quantiles. However, various nonparametric quantile estimators have been proposed in the literature during the recent decades, which likewise may serve in the nonparametric estimation of 2.5% and 97.5% quantiles [9,10,11,12]. In an earlier endeavor, we performed a literature search on nonparametric quantile estimators from which we compared 15 in a simulation study [13]. We assessed the performance of these estimators by the average coverage probability for one newly generated observation from 20,000 fictive trials, and we assumed six different distributions and sample sizes between 30 and 150 to this end. We found that a simple sample quantile estimator based on two rank statistics [9], the Harrell–Davis estimator [14], and the estimators of the Sfakianakis–Verginis type [15] performed, on average, closely to the nominal coverage probability of 95%. The purpose of this paper is to illuminate the variability of the coverage probability of these nonparametric quantile estimators in a simulation study. Three further, possibly promising nonparametric quantile estimators from our former investigation [13] and the classical BA LoAs [3] (as a benchmark measure and for illustrative purposes) are added. In Section 2, all nonparametric quantile estimators and the simulation setup are described. In Section 3, the results are given and illustrated by boxplots. A discussion closes the paper.

## 2. Materials and Methods

The description of the six different methods of nonparametric quantile estimation considered here is kept brief. Further details can be found elsewhere [13].

### 2.1. Nonparametric Quantile Estimators

#### 2.1.1. Sample Quantile Estimator

A random sample of *n* paired differences $X_{1},\ldots X_{n}$ is sorted in increasing order $X_{\left(1\right)}\leq X_{\left(2\right)}\leq \ldots \leq X_{\left(n\right)}$; the symbols here denote the order statistics of the random sample. The SQ estimator is a weighted average of the two order statistics that are closest to including *p* percent of all the observations in the sample:(1)$$\begin{array}{c} SQ=\left(1-\alpha \right)X_{\left(r\right)}+\alpha X_{\left(r+1\right)} \end{array}$$
with α=p(n+1)−r, $r=\left[p(n+1)\right]$, and $\left[x\right]$ is the greatest integer that is less than or equal to *x* [10,12].

#### 2.1.2. Harrell–Davis Estimator

The HD estimator, as well as those estimators that follow, employs linear combinations of all the available order statistics, weighting them according to their relative closeness to the target percentile. They can be given as *L*-statistics, which is, $\sum_{j=1}^{n}W_{j}\cdot X_{\left(j\right)}$, where $W_{j}$ and $X_{\left(j\right)}$ is the weight for the *j*-th order statistic and the *j*-th order statistic itself, respectively [16]. The Harrell–Davis estimator is given by:(2)$$\begin{array}{c} HD=\sum_{i=1}^{n}W_{i}X_{\left(i\right)} \end{array}$$
with weight function:$$W_{i}=I_{i/n}\left\{p(n+1),(1-p)(n+1)\right\}-I_{(i-1)/n}\left\{p(n+1),(1-p)(n+1)\right\},$$
where $I_{i/n}\left\{a,b\right\}$ is the incomplete beta function [9,14].

#### 2.1.3. Bernstein Polynomial Estimator

The BP estimator employs the binomial probability of observing exactly *i* out of *n* events with an event probability of *p*, B(i;n,p), and is given by:
(3)$$\begin{array}{c} BP=\sum_{i=1}^{n}B\left(i-1;n-1,p\right)X_{\left(i\right)} \end{array}$$
according to Cheng [17].

#### 2.1.4. HD Estimator Using a Level Crossing Empirical Distribution

Huang [18] modified the Harrell–Davis estimator (Equation 2) by applying a weighted empirical distribution function instead of the empirical distribution with equal weights 1/n. The Harrell–Davis estimator using a level crossing empirical distribution function can be written as:(4)$$HD_{lc}=\sum_{i=1}^{n}W_{i}X_{\left(i\right)}$$
with weight function:$$W_{i}=I_{q_{i}}\left\{p(n+1),(1-p)(n+1)\right\}-I_{q_{i-1}}\left\{p(n+1),(1-p)(n+1)\right\},$$
the incomplete beta function $I_{q_{i}}\left\{a,b\right\}$, $q_{i}=\sum_{j=1}^{i}w_{j}$, i=1,…,n, and:$$\begin{array}{c} w_{j}=\left\{\begin{array}{cc} \frac{1}{2}\left[1-\frac{n-2}{\sqrt{n(n-1)}}\right] & \;\mathrm{if}\;j=1,n\\ \frac{1}{\sqrt{n(n-1)}} & \;\mathrm{if}\;j=2,3\ldots ,n-1. \end{array}\right. \end{array}$$

#### 2.1.5. Sfakianakis–Verginis Estimator

Sfakianakis and Verginis [15] proposed a group of three estimators, from which we chose the first one due to the similarity of the results in our former investigation [13]. SV estimators are supposed to better estimate quantiles in the tails of a distribution when using small samples, and they employ also the binomial probabilities B(i;n,p) as weights for the ordered statistics $X_{\left(i\right)},i=1,\ldots ,n$:(5)$$\begin{array}{cc} SV & =\frac{2B(0;n,p)+B(1;n,p)}{2}X_{\left(1\right)}+\frac{B(0;n,p)}{2}X_{\left(2\right)}-\frac{B(0;n,p)}{2}X_{\left(3\right)}\\ \; & +\sum_{i=2}^{n-1}\frac{B(i;n,p)+B(i-1;n,p)}{2}X_{\left(i\right)}\\ \; & -\frac{B(n;n,p)}{2}X_{\left(n-2\right)}+\frac{B(n;n,p)}{2}X_{\left(n-1\right)}+\frac{2B(n;n,p)+B(n-1;n,p)}{2}X_{\left(n\right)}. \end{array}$$

#### 2.1.6. Navruz–Özdemir Estimator

Recently, Navruz and Özdemir [19] introduced a new quantile estimator, which is also a linear function of the order statistics with weights of binomial probabilities:(6)$$\begin{array}{cc} NO & =\left(B\left(0;n,p\right)2p+B\left(1;n,p\right)p\right)X_{\left(1\right)}+B\left(0;n,p\right)\left(2-3p\right)X_{\left(2\right)}-B\left(0;n,p\right)\left(1-p\right)X_{\left(3\right)}\\ \; & +\sum_{i=1}^{n-2}\left(B\left(i;n,p\right)\left(1-p\right)+B\left(i+1;n,p\right)p\right)X_{\left(i+1\right)}-B\left(n;n,p\right)pX_{\left(n-2\right)}\\ \; & +B\left(n;n,p\right)\left(3p-1\right)X_{\left(n-1\right)}+\left(B\left(n-1;n,p\right)\left(1-p\right)+B\left(n;n,p\right)\left(2-2p\right)\right)X_{\left(n\right)}. \end{array}$$

### 2.2. Simulation Setup

We assumed six distributions (Figure 1), the choice and number of which were motivated by former simulation studies on nonparametric quantile estimation, i.e., 4–6 symmetric and asymmetric distributions [11,14,15,18,19]:the standard normal distribution;a lognormal distribution with meanlog = 1 and sdlog = 1;a beta distribution with shape parameters α=2 and β=5 (non-centrality parameter λ=0);a beta distribution with shape parameters α=2 and β=2 (λ=0);a chi-squared distribution with 4 degrees of freedom; andan exponential distribution with rate parameter 1.

For sample sizes n=50(50)200,250(250)1000, each distribution, and quantile estimator, we simulated 5000 fictive trials and derived the LoAs. Their respective coverage probability *c* was derived by applying the cumulative distribution function F(x) to the LoAs, i.e.,
(7)c=F(upperLoA)−F(lowerLoA).

This resulted in the distributions of coverage probabilities, which we display with boxplots. We compared the nonparametric quantile estimators with respect to (a) the average (i.e., median and mean) coverage probability and the closeness to the nominal 95% level and (b) their variability in terms of the 5% quantile and the first quartile. The classical BA LoAs served as the benchmark indication of expectable variation under the standard normal distribution; their performance under the remaining distributions illustrated the inappropriateness under non-normal distributions. The Root Mean Squared Error (RMSE) for the estimation of the 2.5% and 97.5% quantiles was supplemented. Here, the true values for these quantiles were known, as were the assumed distributions. Note, though, that the investigation of the coverage probability enabled a simultaneous assessment of both LoAs, whereas the RMSE was related to one of the two LoAs at a time. For instance, underestimation of the lower LoA and underestimation of the upper LoA for one set of estimated LoAs can imply the very same coverage probability as an overestimated lower LoA and an overestimated upper LoA for another set of estimated LoAs. The R code is available as Appendix A (R Version 4.0.2).

## 3. Results

### 3.1. BA LoA

For n=50, the median (mean) coverage probability was close to 0.95 for three distributions, but above the nominal level for the beta distributions (Distributions 3 and 4 in Figure 2; 0.957 (0.953) and 0.972 (0.967), respectively) and slightly below for the exponential distribution (Distribution 6 in Figure 2; 0.943 (0.940)). For Distribution 4, even the first quartile exceeded 0.95 (0.953), indicating overestimation of the BA LoA, which is visible by the box lying completely above 0.95 in Figure 2. The same applied in the case of Distribution 3 for sample sizes of n≥200 and for Distribution 2 for sample sizes of n≥500 (Figure 2 and Figure 3). In the case of Distribution 6, the median (mean) coverage probability fell short of the nominal level for all sample sizes, even for n=1000 (0.948 (0.948)).

### 3.2. Nonparametric Quantile Estimators

#### 3.2.1. SQ Estimator

The median (mean) coverage probability of the simple sample quantile estimator SQ was 0.958–0.959 (0.952–0.954) for n=50; the first quartile was 0.938–0.939; and the third quartile was 0.972–0.975 (Figure 4). For larger sample sizes (Appendix A), the coverage probability converged to the nominal level, always exceeding 0.95 on average (e.g., n=150: median 0.952–0.953 (mean 0.950–0.951), first quartile 0.939–0.94). The 5% quantile increased likewise from 0.896 (n=50) to 0.938 (n=1000); see Table 1. A distinguishing feature of SQ was the similarity of the boxplots across both distributions and sample sizes (Figure 4, Appendix A).

#### 3.2.2. HD Estimator

The median (mean) coverage probability of the HD estimator was 0.941–0.946 (0.936–0.941) for n=50; the first quartile was 0.920–0.924; and the third quartile was 0.958–0.964 (Figure 5). For larger sample sizes (Appendix A), the coverage probability converged, on average, to the nominal level (e.g., n=150: median 0.949–0.952 (mean 0.947–0.950), first quartile 0.938–0.94). The 5% quantile was slightly below that of SQ for n=50,100 (Table 1), but identical for n≥150. The HD estimator performed very similar to the SQ estimator for n≥150 (Figure 5, Appendix A).

#### 3.2.3. BP Estimator

The BP estimator fell short of the nominal coverage probability for n≤250 (Appendix A). For n=250, for instance, the median (mean) and first quartile coverage probability was 0.945–0.947 (0.944–0.946) and 0.936–0.938, respectively. The median (mean) coverage probability reached 0.949 (0.948–0.949) for n=750,1000.

Comparing the HD and BP estimator for n=200 in terms of the RMSE, the RMSE was smaller for the HD than for the BP estimator for four distributions in the estimation of the 2.5% percentile (Table 2) and for one distribution in the estimation of the 97.5% percentile (Table 3). For three distributions, the RMSE was smaller for the BP than for the HD estimator in the estimation of the 97.5% percentile.

In general, all nonparametric LoAs performed considerably close to each other in the estimation of the 2.5% percentile for sample sizes of n≥100 (Table 2). In the estimation of the 97.5% percentile, this was the case for n≥200 (Table 3).

#### 3.2.4. HD lc Estimator

The HD lc estimator performed quite similar to the BP estimator (Appendix A). The median (mean) coverage probability reached, though, 0.949–0.95 (0.948–0.949) for n=500; for n=250, the median (mean) and first quartile coverage probability was 0.947–0.948 (0.946–0.948) and 0.938–0.94, respectively.

#### 3.2.5. SV Estimator

The median (mean) coverage probability of the SV estimator was 0.942–0.946 (0.937–0.941) for n=50; the first quartile was 0.920–0.924; and the third quartile was 0.96–0.963 (Figure 6). For larger sample sizes (Appendix A), the coverage probability converged, on average, to the nominal level (e.g., n=150: median 0.95–0.953 (mean 0.948–0.951), first quartile 0.938–0.942). The 5% quantile was slightly below that of SQ for n=50,100 (Table 1), but identical for n≥150. The SV estimator performed very similar to the SQ estimator for n≥150 and, in general, close to identical to the HD estimator with respect to the 5% quantile, the first quartile, and the median of the coverage probability for all sample sizes (±0.001).

#### 3.2.6. NO Estimator

The NO estimator achieved a median (mean) and first quartile coverage probability of 0.95–0.951 (0.949–0.950) and 0.944–0.945 for n=500, but fell short of the nominal coverage probability of 0.95 for smaller sample sizes (Appendix A). Only for sample sizes of n=500,750,1000, the NO estimator performed comparably to the SQ estimator.

## 4. Discussion

### 4.1. Key Finding

The SQ estimator, a simple sample quantile estimator based on two rank statistics, conservatively kept, on average, the nominal coverage level of 95% for n=50 and converged rapidly towards it with increasing sample size. The variability of the SQ estimator, measured in terms of the 5% quantile and the first quartile of the coverage probability, was smallest amongst all considered nonparametric quantile estimators for n=50,100. For sample sizes of n≥150, both the HD and the SV estimator performed likewise well in terms of average performance and small variability. The remaining nonparametric estimators considered in the study did so for larger sample sizes only (BP: n≥750, HD_*lc*_: n≥500, and NO: n≥500).

### 4.2. What Does This Add to What Is Known

Harrell and Davis [14] investigated the mean squared error of their estimator to measure its efficiency with sample sizes of a maximum n=60 and did not recommend the HD estimator for small *n* and extreme *p*. Dielman, Lowry, and Pfaffenberger [12] compared several quantile estimators in terms of bias with sample sizes n=10,15,25,30 and *p* as small as 0.02. They concluded that there was not one best estimator across scenarios, but the HD estimator performed well in a wide range of cases, except when p=0.02,0.98. They highlighted that the HD estimator makes use of many data points and, therefore, performed better in the estimation of middle quantiles. On the contrary, sample quantile estimators, like the SQ estimator, performed well at p=0.02, and the HD estimator performed better at larger sample sizes. Sfakianakis and Verginis [15] compared the SV estimator with the HD and SQ estimators in terms of bias and mean squared error for p=0.01,0.05(0.05),0.95,0.99 and *n* = 5–750. They concluded that the SV estimator performed better than the HD and SQ estimators in most of the examined cases did.

The nonparametric estimation of the LoAs involves two tail percentiles, p=0.025,0.975, which need to be considered simultaneously. Earlier, we indicated that the SQ estimator outperformed other quantile estimators for n=50 in terms of the mean coverage probability for the next observation, but the HD estimator and estimators of the Sfakianakis–Verginis type performed likewise well for sample sizes exceeding 80 observations [13]. Here, we investigated both the average performance and variability of nonparametric quantile estimators for the LoA. The SQ estimator outperformed the HD and SV estimators for n=50 and was slightly better for n=100, whereas the SQ, HD, and SV estimators performed identically well for n≥150. The similarity of the boxplots for the SQ estimator across both distributions and sample sizes was striking.

### 4.3. What Is the Implication, and What Should Change Now?

Whenever violated assumptions for the paired differences prohibit the immediate derivation of the classical BA LoA and neither an appropriate preparatory transformation nor a regression approach are practicable alternatives [3,8,20,21,22], the SQ estimator may serve as a basis for the derivation of nonparametric LoAs. For sample sizes n≥150, this holds also true for the HD and SV estimators, although all available differences contribute to the quantile estimation, and the target quantiles are extreme tail percentiles. The SQ estimator is not defined for n<40; however, a reasonably sized method comparison study is likely to enroll at least 40 subjects anyway [23,24,25,26]. Finally, the 95% confidence intervals should supplement nonparametric LoAs to indicate the estimates’ uncertainty [27,28]. Bootstrapping techniques may serve this purpose for the SQ estimator.

## Figures and Tables

**Figure 1 ijerph-17-08330-f001:**
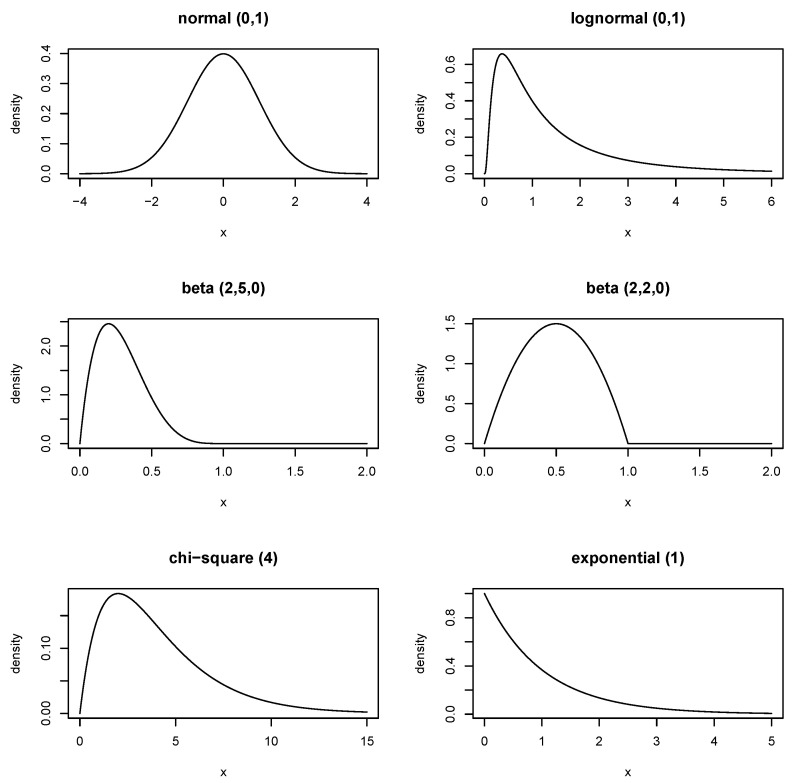
Density functions of the assumed distributions.

**Figure 2 ijerph-17-08330-f002:**
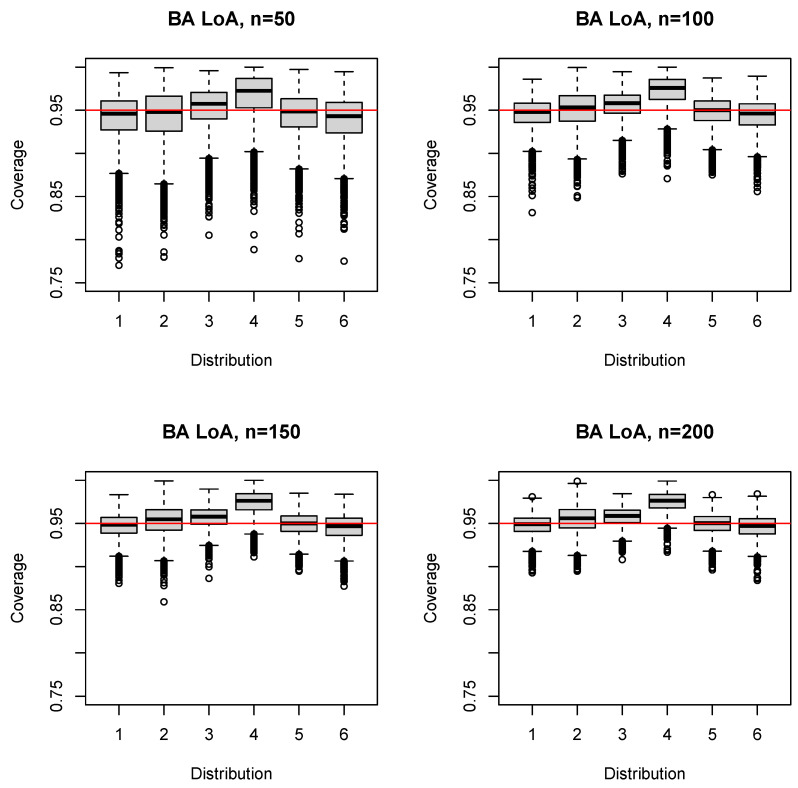
Boxplots of the coverage probability for the BA LoA and n=50(50)200.

**Figure 3 ijerph-17-08330-f003:**
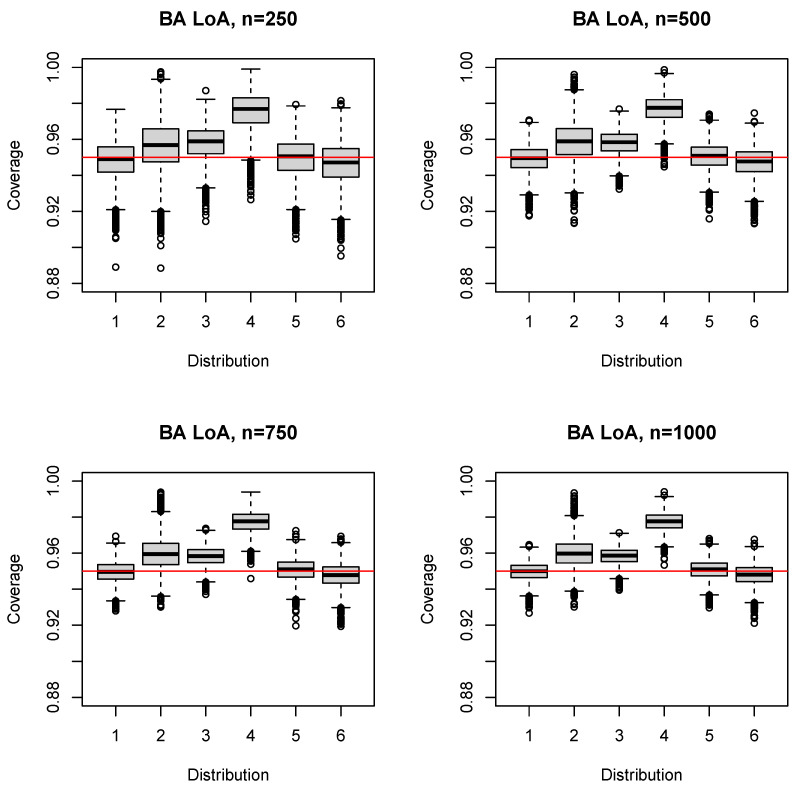
Boxplots of the coverage probability for the BA LoA and n=250(250)1000.

**Figure 4 ijerph-17-08330-f004:**
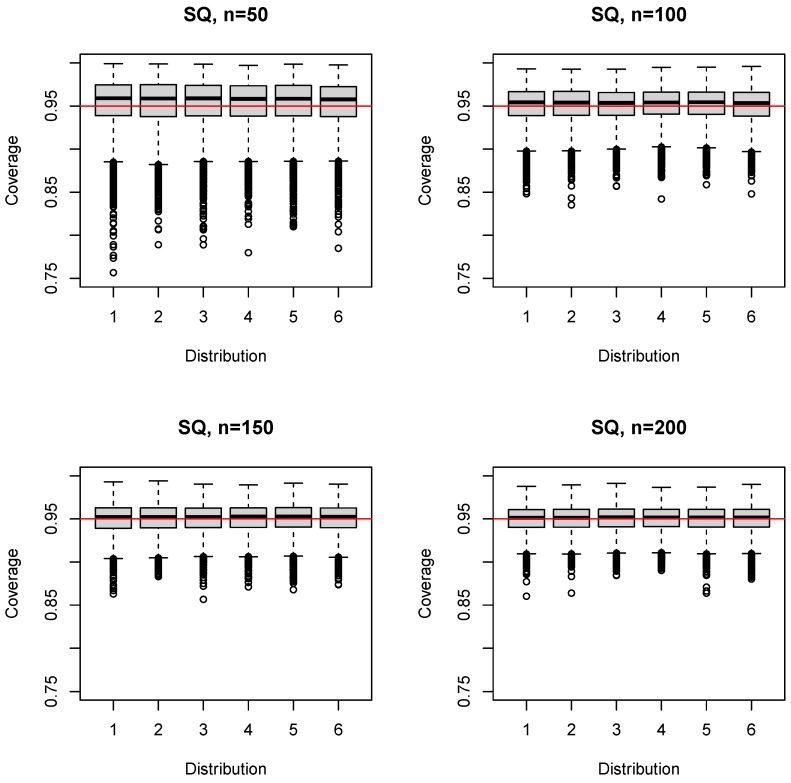
Boxplots of the coverage probability for the SQ estimator and n=50(50)200.

**Figure 5 ijerph-17-08330-f005:**
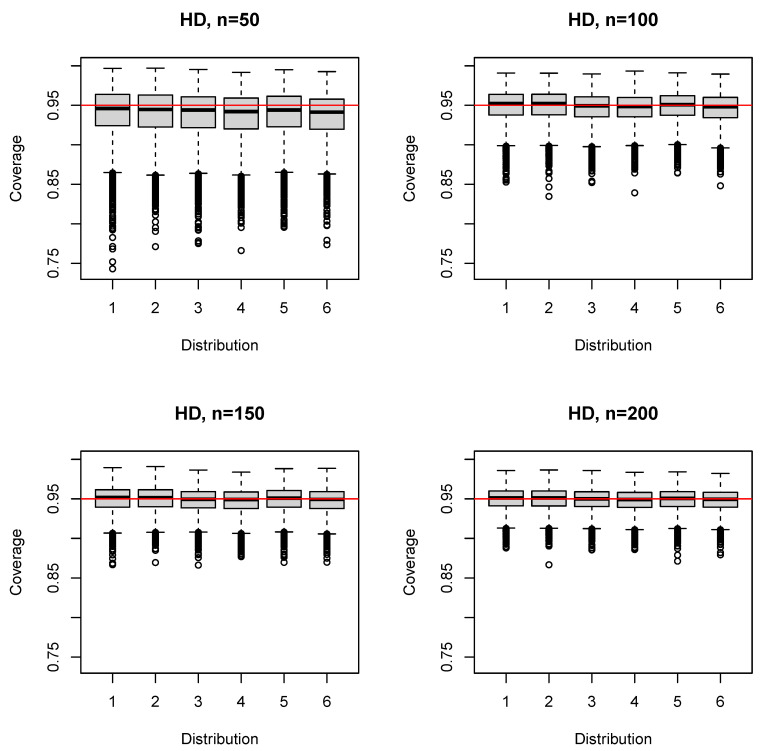
Boxplots of the coverage probability for the HD estimator and n=50(50)200.

**Figure 6 ijerph-17-08330-f006:**
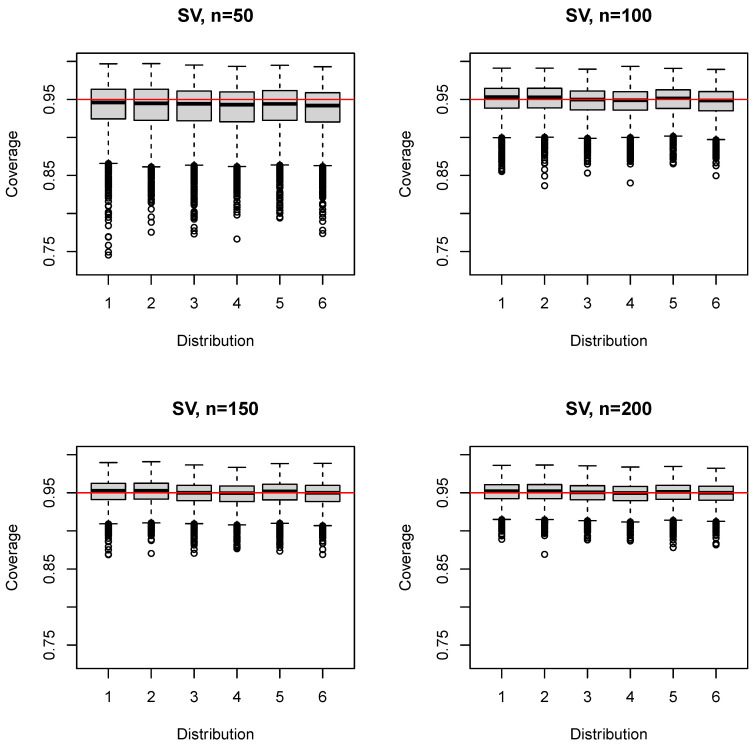
Boxplots of the coverage probability for the SV estimator and n=50(50)200.

**Table 1 ijerph-17-08330-t001:** Five percent quantiles of the coverage probability by the estimator and sample size *n*. For BA LoA, the 5% quantiles are shown assuming a standard normal distribution (benchmark values); for the nonparametric estimators, the minimum of the 5% quantiles across all 6 distributions are shown.

Estimator	50	100	150	200	250	500	750	1000
BA LoA	0.893	0.914	0.922	0.927	0.929	0.936	0.939	0.941
SQ	0.896	0.913	0.919	0.922	0.925	0.933	0.936	0.938
HD	0.880	0.910	0.919	0.922	0.926	0.933	0.936	0.938
BP	0.859	0.897	0.911	0.917	0.921	0.931	0.935	0.937
HD_*lc*_	0.871	0.903	0.914	0.920	0.923	0.932	0.936	0.938
SV	0.880	0.911	0.920	0.923	0.926	0.933	0.936	0.939
NO	0.491	0.728	0.858	0.903	0.920	0.934	0.936	0.938

**Table 2 ijerph-17-08330-t002:** RMSE of the estimated 2.5% quantiles by sample size, estimator, and distribution.

*n*	Estimator	Normal (0,1)	Lognormal (0,1)	Beta (2,5,0)	Beta (2,2,0)	Chi-Squared (4)	Exponential (1)
50	BA LoA	0.20	2.34	0.070	0.045	1.97	0.94
	SQ	0.34	0.040	0.015	0.031	0.17	0.015
	HD	0.28	0.037	0.014	0.028	0.16	0.017
	BP	0.26	0.043	0.016	0.032	0.18	0.023
	HD lc	0.26	0.039	0.014	0.029	0.17	0.019
	SV	0.27	0.038	0.014	0.029	0.16	0.017
	NO	0.28	0.049	0.018	0.037	0.21	0.024
100	BA LoA	0.14	2.48	0.070	0.038	2.00	0.96
	SQ	0.22	0.029	0.011	0.023	0.124	0.012
	HD	0.20	0.026	0.010	0.021	0.11	0.012
	BP	0.18	0.028	0.010	0.022	0.12	0.014
	HD lc	0.19	0.027	0.010	0.021	0.12	0.013
	SV	0.20	0.026	0.009	0.020	0.11	0.012
	NO	0.19	0.029	0.011	0.022	0.13	0.014
150	BA LoA	0.11	2.52	0.070	0.035	2.01	0.97
	SQ	0.18	0.024	0.009	0.019	0.104	0.010
	HD	0.16	0.022	0.008	0.017	0.093	0.009
	BP	0.15	0.023	0.009	0.018	0.098	0.011
	HD lc	0.16	0.022	0.008	0.017	0.095	0.010
	SV	0.16	0.022	0.008	0.017	0.092	0.009
	NO	0.15	0.023	0.009	0.018	0.099	0.011
200	BA LoA	0.095	2.55	0.070	0.034	2.01	0.98
	SQ	0.16	0.021	0.008	0.017	0.092	0.009
	HD	0.14	0.019	0.007	0.015	0.082	0.008
	BP	0.14	0.020	0.007	0.016	0.085	0.009
	HD lc	0.14	0.019	0.007	0.015	0.083	0.008
	SV	0.14	0.019	0.007	0.015	0.081	0.008
	NO	0.14	0.020	0.008	0.016	0.085	0.009
250	BA LoA	0.086	2.59	0.071	0.034	2.02	0.98
	SQ	0.14	0.019	0.007	0.015	0.080	0.008
	HD	0.13	0.017	0.006	0.014	0.073	0.007
	BP	0.12	0.018	0.007	0.014	0.074	0.008
	HD lc	0.12	0.017	0.007	0.014	0.073	0.008
	SV	0.13	0.017	0.006	0.014	0.072	0.007
	NO	0.12	0.018	0.007	0.014	0.075	0.008
500	BA LoA	0.062	2.65	0.070	0.033	2.03	0.98
	SQ	0.096	0.013	0.005	0.011	0.059	0.006
	HD	0.091	0.012	0.005	0.010	0.055	0.005
	BP	0.090	0.012	0.005	0.010	0.056	0.006
	HD lc	0.090	0.012	0.005	0.010	0.056	0.006
	SV	0.091	0.012	0.005	0.010	0.055	0.005
	NO	0.090	0.012	0.005	0.010	0.056	0.006
750	BA LoA	0.050	2.68	0.071	0.033	2.02	0.98
	SQ	0.077	0.011	0.004	0.009	0.048	0.005
	HD	0.073	0.010	0.004	0.009	0.045	0.004
	BP	0.073	0.010	0.004	0.009	0.046	0.005
	HD lc	0.073	0.010	0.004	0.009	0.046	0.004
	SV	0.073	0.010	0.004	0.009	0.045	0.004
	NO	0.073	0.010	0.004	0.009	0.046	0.005
1000	BA LoA	0.043	2.68	0.071	0.033	2.02	0.99
	SQ	0.068	0.010	0.004	0.008	0.042	0.004
	HD	0.065	0.009	0.003	0.007	0.040	0.004
	BP	0.064	0.009	0.004	0.007	0.040	0.004
	HD lc	0.064	0.009	0.003	0.007	0.040	0.004
	SV	0.065	0.009	0.003	0.007	0.040	0.004
	NO	0.064	0.009	0.003	0.007	0.040	0.004

**Table 3 ijerph-17-08330-t003:** RMSE of the estimated 97.5% quantiles by sample size, estimator, and distribution.

*n*	Estimator	Normal (0,1)	Lognormal (0,1)	Beta (2,5,0)	Beta (2,2,0)	Chi-Squared (4)	Exponential (1)
50	BA LoA	0.20	2.16	0.052	0.046	1.75	0.81
	SQ	0.34	3.40	0.059	0.032	2.06	0.87
	HD	0.28	2.60	0.049	0.029	1.62	0.68
	BP	0.26	2.01	0.048	0.033	1.40	0.59
	HD lc	0.26	2.30	0.047	0.030	1.49	0.63
	SV	0.28	2.54	0.049	0.029	1.61	0.68
	NO	0.73	2.81	0.23	0.33	4.05	1.38
100	BA LoA	0.14	1.83	0.046	0.039	1.63	0.76
	SQ	0.22	1.85	0.039	0.023	1.27	0.54
	HD	0.20	1.87	0.034	0.020	1.16	0.49
	BP	0.19	1.47	0.033	0.022	1.02	0.43
	HD lc	0.19	1.60	0.033	0.021	1.07	0.45
	SV	0.20	1.95	0.034	0.020	1.18	0.50
	NO	0.38	1.62	0.13	0.18	2.05	0.70
150	BA LoA	0.11	1.69	0.044	0.036	1.63	0.74
	SQ	0.18	1.37	0.031	0.019	0.99	0.43
	HD	0.16	1.36	0.028	0.017	0.92	0.40
	BP	0.16	1.17	0.027	0.018	0.84	0.36
	HD lc	0.16	1.23	0.028	0.017	0.87	0.38
	SV	0.16	1.45	0.028	0.017	0.94	0.41
	NO	0.18	1.24	0.048	0.074	0.99	0.39
200	BA LoA	0.096	1.60	0.043	0.034	1.62	0.74
	SQ	0.16	1.18	0.028	0.017	0.87	0.37
	HD	0.14	1.14	0.025	0.015	0.79	0.34
	BP	0.14	1.02	0.025	0.016	0.75	0.32
	HD lc	0.14	1.07	0.025	0.016	0.77	0.33
	SV	0.14	1.19	0.025	0.015	0.80	0.34
	NO	0.14	1.18	0.026	0.026	0.79	0.34
250	BA LoA	0.088	1.55	0.042	0.033	1.62	0.74
	SQ	0.14	1.02	0.024	0.015	0.76	0.33
	HD	0.13	0.99	0.023	0.014	0.71	0.30
	BP	0.13	0.91	0.022	0.014	0.68	0.29
	HD lc	0.13	0.94	0.022	0.014	0.69	0.30
	SV	0.13	1.02	0.022	0.014	0.71	0.31
	NO	0.13	1.10	0.022	0.014	0.74	0.32
500	BA LoA	0.062	1.39	0.043	0.032	1.61	0.73
	SQ	0.097	0.69	0.018	0.011	0.53	0.22
	HD	0.091	0.67	0.017	0.010	0.50	0.21
	BP	0.090	0.64	0.016	0.010	0.49	0.21
	HD lc	0.090	0.65	0.016	0.010	0.49	0.21
	SV	0.091	0.67	0.017	0.010	0.50	0.21
	NO	0.094	0.72	0.017	0.010	0.53	0.22
750	BA LoA	0.050	1.34	0.043	0.033	1.60	0.73
	SQ	0.079	0.57	0.014	0.009	0.44	0.18
	HD	0.075	0.55	0.014	0.008	0.42	0.17
	BP	0.074	0.53	0.014	0.008	0.41	0.17
	HD lc	0.075	0.54	0.014	0.008	0.41	0.17
	SV	0.075	0.55	0.014	0.008	0.42	0.17
	NO	0.076	0.57	0.014	0.008	0.43	0.18
1000	BA LoA	0.043	1.31	0.042	0.032	1.60	0.73
	SQ	0.068	0.48	0.012	0.008	0.37	0.16
	HD	0.065	0.47	0.012	0.007	0.36	0.15
	BP	0.064	0.46	0.012	0.007	0.35	0.15
	HD lc	0.065	0.46	0.012	0.007	0.35	0.15
	SV	0.065	0.47	0.012	0.007	0.36	0.15
	NO	0.066	0.48	0.012	0.007	0.36	0.16

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
