# Peer review of "Nonparametric Limits of Agreement in Method Comparison Studies: A Simulation Study on Extreme Quantile Estimation"

_ijerph, 2020, doi:10.3390/ijerph17228330_

Round 1
Reviewer 1 Report
In the context of comparison studies on quantitative outcomes, the author presents a simulation study in order to investigated the variability of the coverage probability of six estimators for the 2.5% and 97.5% quantiles of the differences: the sample quantile estimator (SQ), the Harrell-Davis estimator (HD), the Bernstein Polynomial estimator, the Harrell-Davis estimator using a level crossing empirical distribution, the Sfakianakis-Verginis (SV) estimator and the Navruz-Özdemir estimator.
This study presents itself as an extension of a previous work by the author where it was concluded that simple sample quantil estimator based on two rank statistics, the HF estimator, and the SV estimator performed, on average, closely to the nominal coverage probability of 95%. The only novelty is the fact that the author, throught a simulation study shows the variability of the coverage probability of the six nonparametric quantile estimators with n = 50(50)200, 250(250)1000. Concluding that the SQ estimator outperformed the HD and SV estimator for n = 50 and was slightly better for n = 100, whereas the SQ, HD, and SV estimator performed identically well for bigger sample sizes.
The article is very well written, and the subject is exposed in a clear and well organized way.
Since the innovative aspect of the article relies on the simulation study, I have a concern related to the reduced number of simulated distributions and variation of the parameters for a more robust generalization of the results. Also, it would be advantageous for the article to clarify the motivation behind the choice of distributions and their parameters.
Author Response
Dear Reviewer 1.
Thank you very much for your comments and suggestions. Please find below your previous input and my answer.
In the context of comparison studies on quantitative outcomes, the author presents a simulation study in order to investigated the variability of the coverage probability of six estimators for the 2.5% and 97.5% quantiles of the differences: the sample quantile estimator (SQ), the Harrell-Davis estimator (HD), the Bernstein Polynomial estimator, the Harrell-Davis estimator using a level crossing empirical distribution, the Sfakianakis-Verginis (SV) estimator and the Navruz-Özdemir estimator.
This study presents itself as an extension of a previous work by the author where it was concluded that simple sample quantil estimator based on two rank statistics, the HF estimator, and the SV estimator performed, on average, closely to the nominal coverage probability of 95%. The only novelty is the fact that the author, throught a simulation study shows the variability of the coverage probability of the six nonparametric quantile estimators with n = 50(50)200, 250(250)1000. Concluding that the SQ estimator outperformed the HD and SV estimator for n = 50 and was slightly better for n = 100, whereas the SQ, HD, and SV estimator performed identically well for bigger sample sizes.
The article is very well written, and the subject is exposed in a clear and well organized way.
Since the innovative aspect of the article relies on the simulation study, I have a concern related to the reduced number of simulated distributions and variation of the parameters for a more robust generalization of the results. Also, it would be advantageous for the article to clarify the motivation behind the choice of distributions and their parameters.
Answer: I agree that more scenarios would extend the generalizability of the results, but probably only to a limited extent. Most simulation studies on quantile estimation employed 4-6 distributions, comprising both symmetric and asymmetric ones. I have added a note on this at the beginning of Section 2.2 (Simulation setup), including references.
Reviewer 2 Report
See the PDF.

Reviewer 3 Report
The author considered different non-parametric estimators to obtain the Bland-Altman Limits of Agreement (LoA). In the end, it is showed that the SQ estimator may serve as a basis for the derivation of nonparametric LoA. I believe that the manuscript provides interesting results and may be of interest to a wide readership.
However, I am wondering why the author only presented the coverage probabilities for the estimators, I would like to see the average Bias of the estimators and the mean square errors, these metrics are very important when comparing different methods.
Author Response
Dear Reviewer 3.
Thank you very much for your comments and suggestions. Please find below your previous input and my answer.
The author considered different non-parametric estimators to obtain the Bland-Altman Limits of Agreement (LoA). In the end, it is showed that the SQ estimator may serve as a basis for the derivation of nonparametric LoA. I believe that the manuscript provides interesting results and may be of interest to a wide readership.
However, I am wondering why the author only presented the coverage probabilities for the estimators, I would like to see the average Bias of the estimators and the mean square errors, these metrics are very important when comparing different methods.
Answer: I have added root mean squared errors for the estimation of both the 2.5% and the 97.5% percentile (new Tables 2 and 3). However, I am still convinced that the investigation of the coverage probability for nonparametric Limits of Agreement prevails over RMSE as RMSE cannot assess both target percentiles simultaneously (see also the final of Section 2.2 (Simulation setup)).
Round 2
Reviewer 3 Report
Although the author has no included the suggestion. In my opinion, the manuscript can be accepted in this current form.